# Assessment of Fermented Soybean Meal on *Salmonella typhimurium* Infection in Neonatal Turkey Poults

**DOI:** 10.3390/ani10101849

**Published:** 2020-10-11

**Authors:** Arantxa Morales-Mena, Sergio Martínez-González, Kyle D. Teague, Lucas E. Graham, Roberto Señas-Cuesta, Christine N. Vuong, Howard Lester, Daniel Hernandez-Patlan, Bruno Solis-Cruz, Benjamin Fuente-Martinez, Xochitl Hernandez-Velasco, Billy M. Hargis, Guillermo Tellez-Isaias

**Affiliations:** 1Centro de Ensenanza, Investigacion y Extension en Produccion Avicola, Facultad de Medicina Veterinaria y Zootecnia (FMVZ), Universidad Nacional Autonoma de Mexico, Ciudad de Mexico 13209, Mexico; arantxa.mo.me@gmail.com (A.M.-M.); benjaminfuente@yahoo.com.mx (B.F.-M.); 2Unidad Académica de Medicina Veterinaria y Zootecnia, Universidad Autónoma de Nayarit, Compostela, Nayarit 63700, Mexico; sergiotepic@hotmail.com; 3Department of Poultry Science, University of Arkansas, Fayetteville, AR 72701, USA; kdteague@email.uark.edu (K.D.T.); leg004@email.uark.edu (L.E.G.); rsenascu@uark.edu (R.S.-C.); vuong@uark.edu (C.N.V.); hlester@uark.edu (H.L.); bhargis@uark.edu (B.M.H.); 4Laboratorio 5: LEDEFAR, Unidad de Investigacion Multidisciplinaria, Facultad de Estudios Superiores (FES) Cuautitlan, Universidad Nacional Autonoma de Mexico (UNAM), Cuautitlan Izcalli 54714, Mexico; danielpatlan@comunidad.unam.mx (D.H.-P.); bruno_sc@comunidad.unam.mx (B.S.-C.); 5Departamento de Medicina y Zootecnia de Aves, FMVZ, Universidad Nacional Autonoma de Mexico, Ciudad de Mexico 04510, Mexico; xochitl_h@yahoo.com

**Keywords:** intestinal permeability, fermented soybean meal, probiotic, turkey poults, *Salmonella typhimurium*

## Abstract

**Simple Summary:**

Fermented soybean meal (FSBM) improved the performance of neonatal turkey poults, and these changes were associated with morphometric changes in the intestinal mucosa, as well as reduction of leaky gut, within turkeys challenged with *Salmonella typhimurium* (ST) in two experimental infective models. Although the two strains used for the fermentation process showed in vitro activity against ST, no significant effect was observed in vivo. Nevertheless, FSBM reduced the severity of the ST infection in the therapeutic model. The fermentation with different beneficial bacteria and different inclusion rates of FSBM requires further investigation.

**Abstract:**

This study’s objective was to evaluate the effect of the fermented soybean meal (FSBM) on *Salmonella typhimurium* (ST) to turkey poults using two models of infection. In the prophylactic model, one-day-old turkeys were randomly allocated to one of four different groups (*n* = 30 turkeys/group): (1) Control group, (2) FSBM group, (3) Control group challenged with ST (Control + ST), and (4) FSBM group challenged with ST (FSBM + ST). On day 9 of age, all poults were orally challenged with 10^6^ colony forming units (CFU) ST and 24 h post-inoculation, intestinal samples were collected to determine ST recovery and morphometric analysis. Blood samples were collected to evaluate serum fluorescein isothiocyanate-dextran (FITC-d). In the therapeutic model, a similar experimental design was used, but turkeys were orally gavaged 10^4^ CFU ST on day 1, and samples were collected at day 7. FSBM improved performance and reduced leaky gut in both experimental infective models. In the prophylactic model, FSBB induced morphology changes in the mucosa. Although the strains (*Lactobacillus salivarius* and *Bacillus licheniformis*) used for the fermentation process showed in vitro activity against ST, no significant effect was observed in vivo. The fermentation with different beneficial bacteria and different inclusion rates of FSBM requires further investigation.

## 1. Introduction

Food-borne or water-borne microbial pathogens are associated with diarrheal disorders killing an estimated two million people annually at the global level [1]. Just in the United States of America, it has been estimated that nontyphoidal *Salmonella* causes over one million foodborne infections every year [2]. Several multistate outbreaks of human *Salmonella* infections have been associated with the consumption of poultry products [3]. On the other hand, there is a public and scientific concern of the continuing emergence of microbial resistance to antibiotics due to the widespread use of antibiotics in farm animal production [4]. Probiotics have become a successful alternative to reduce the extensive use of antibiotics and control foodborne pathogens, such as *Salmonella* spp. [5]. However, in recent years, the utilization of probiotic bacteria has also been used to reduce the anti-nutritional factors of soybean meal through fermentation methods. Several studies have demonstrated that the fermentation process increases the nutritional value [6], reduces intestinal inflammation [7], and minimizes colonization of enteric pathogens [8,9]. In the present study, a *Lactobacillus salivarius* and a *Bacillus licheniformis* strains isolated from poultry were used to ferment soybean meal. Both strains exhibited in vitro properties capable of reducing *Salmonella typhimurium* (ST). Hence, it was postulated that the fermented soybean meal (FSBM) would improve performance, morphometric changes in the intestinal mucosa, reduced intestinal permeability, and provide protection against ST to neonatal turkey poults using two published models of infection [10].

## 2. Materials and Methods

### 2.1. Isolation and Selection of Inoculum Bacteria Used for Fermentation

Bacteria used for the inoculum were previously isolated from the cecal content of two 5-week commercial broiler chickens. Cecal and ileum (Meckel’s diverticulum to cecal tonsils) content from these birds was collected, and both sections were flushed with PBS. Epithelium and intestinal contents were homogenized, serially diluted, and plated on de Man Rogosa Sharpe (MRS) agar plates (Catalog no. 288110, Becton Dickinson and Co., Sparks, MD, USA) or tryptic soy (TS) agar plates (Catalog no. 211822, Becton Dickinson, Sparks, MD, USA) respectively to obtain one pure colony from each sample. The *Bacillus* isolate was selected as a direct-fed microbial (DFM) candidate in previous experiments based on its ability for in vitro degradation of carbohydrates found in soybean meal (SBM) that are poorly digested by poultry [11]. Both isolates were identified by 16S rRNA sequence analyses (Microbial ID Inc., Newark, DE, USA). The report showed that the strains were *Lactobacillus salivarius* (LS) and *Bacillus licheniformis* (BL). Aliquots of these isolates were made and used in the present study.

### 2.2. Salmonella typhimurium Strain and Culture Conditions

The challenge organism used in all experiments was a poultry isolate of *Salmonella enterica* serovar *Typhimurium* (ST) obtained from the United States Department of Agriculture (USDA) National Veterinary Services Laboratory (Ames, IA, USA). This strain is resistant to 25 µg/mL of novobiocin (NO, catalog no. N-1628, Sigma, St. Louis, MO, USA) and is selected for resistance to 20 µg/mL of nalidixic acid (NA, catalog no. N-4382, Sigma). Final concentrations of ST used for challenge were further verified by serial dilution and plating on brilliant green agar (BGA, Catalog no. 70134, Sigma, St. Louis, MO, USA) with NO and NA for enumeration in each experiment.

### 2.3. In Vitro Assessment of Antimicrobial Activity against Salmonella Enterica Serovar typhimurium

The LS and BL isolates were screened for in vitro antimicrobial activity against ST, as described by Yang [12]. Both LS and BL isolates exhibited in vitro antimicrobial activity against ST. Ten microliters of each isolate were placed in the center of MRS or TSA plates, correspondingly. After 24 h of incubation at 37 °C, the plated samples were overlaid with TSA (Tryptic Soy Agar, catalog no. 211822, Becton Dickinson, Sparks, MD, USA) containing 10^6^ cfu/mL of ST. After 24 h of incubation at 37 °C, a zone of inhibition was observed in each plate, respectively.

### 2.4. Preparation of Fermented Soybean Meal

Before the fermentation process, microbiological analysis of the SBM was revealed to contain 1 × 10^3^ colony forming units (CFU)/g of total lactic acid bacteria (LAB), and 1 × 10^4^ CFU/g of total aerobic bacteria. A sample of the SBM was pasteurized at 70 °C for 10 min to eliminate vegetative cells and validate the number of spores from aerobic bacteria per gram on samples plated on TS agar. The results showed that the SBM contained 1 × 10^2^ spores/g. The fermented soybean meal (FSBM) was produced through fermentation of the soybean meal with *L. salivarius* and *B. licheniformis* according to methods presented in a previous study by Jazi et al. [13]. For the fermentation process, one kg of SBM was inoculated with the two-culture media and the two bacteria at the same time. Three liters of MRS containing 2 × 10^9^ CFU/mL *L. salivarius* and three liters of TS broth containing 3 × 10^8^ CFU/mL *B. licheniformis* and incubated for 24 h at 37 °C, followed by additional 24 h at 30 °C. FSBM was then dried in a hot-air oven at 60 °C for 26 h. Following fermentation, samples from the FSBM were collected for determination of total LAB or total aerobic bacteria. A total of 2 × 10^5^ LAB/g and 5 × 10^4^ total aerobic bacteria were recovered from the FSBM. Pasteurization of the FSBM showed a total of 3 × 10^3^ spores/g.

### 2.5. Experimental Diets

All the experimental diets used in this study were formulated to approximate the nutritional requirements of turkey poults [14] and adjusted to the feeding guidelines for Nicholas Turkeys (Avigen, Turkesy) commercial recommendations. No antibiotics were added to the diet. A basal diet was formulated with SBM for the control group diets, and for the treatment group diets, the FSBM replaced 100% of the SBM. Diet samples were chemically analyzed, amino acids were determined by ion chromatography (Biochrom Ltd., Cambridge, UK) (Table 1). Dry matter content of SBM and FSBM was 89.3% and 88.38%, respectively. Interestingly, FSBM did not affect the nutritional value of the diet. In the in vivo experiments, all animal handling procedures were in compliance with the Institutional Animal Care and Use Committee at the University of Arkansas under permit number 18029.

### 2.6. In Vitro Digestion Model, Experiment 1

In this experiment, the antimicrobial activity of FSBM against ST was determined using an in vitro digestion model previously described that simulates the pH and enzymatic conditions present in the crop, proventriculus, and intestine of broilers [15]. Briefly, for all the gastrointestinal compartments simulated during the in vitro digestion model, a BOD incubator (Biochemical oxygen demand incubator, model 2020, VWR, Houston, TX, USA) customized with an orbital shaker (Standard orbital shaker, model 3500, VWR, Houston, TX, USA) was used for mixing the feed content in the experimental tubes at 19 rpm. Additionally, all tube samples were held in a 30 degrees inclination position to facilitate the proper blending of feed particles and the enzyme solutions incorporated throughout the assay. The first gastrointestinal compartment simulated was the crop, where 5 g of feed and 10 mL of 0.03 M hydrochloric acid (HCL, catalog no. HX0607-2, EMD Millipore corporation, Billerica, MA, USA) were placed in 50 mL polypropylene centrifuge tubes and mixed vigorously reaching a pH value around 5.20, next the tubes were incubated for 30 min. The second gastrointestinal compartment simulated was the proventriculus, where 3000 U of pepsin per g of feed where used (catalog no. P700, Sigma-Aldrich, St. Louis, MO, USA) and 2.5 mL of 1.5 M HCl were added to each of the tubes, reaching a pH between 1.4 to 2.00, then all tubes were incubated for 45 min. The third and final gastrointestinal compartment simulated was the intestinal section. In this case, 6.84 mg of 8 x pancreatin (catalog no. P7545, Sigma-Aldrich, St. Louis, MO, USA) were used per g of feed and included in 6.5 mL of 1.0 M sodium bicarbonate (NaHCO_3_, catalog no. S6014, Sigma-Aldrich, St. Louis, MO, USA), the pH ranged between 6.4 and 6.8, and all tube samples were incubated for 2 h. The complete in vitro digestion process took 3 h and 15 min.

### 2.7. Effect of Prophylactic Administration of FSBM on Salmonella typhimurium, Experiment 2

This experiment was conducted to evaluate the prophylactic administration of FSBM in reducing the incidence of ST in turkey poults. One-day-old female turkey poults were obtained from a local hatchery, neck tagged, and randomly allocated to one of four different groups (*n* = 30 turkeys/group): (1) Control group, (2) FSBM group, (3) Control group challenged with ST (Control + ST), and (4) FSBM group challenged with ST (FSBM + ST). Poults were individually weighed at reception and on day 9 to obtain body weight (BW) and body weight gain (BWG). Upon arrival, ten extra day-of-hatch poults were euthanized by CO_2_ inhalation. Ceca-cecal tonsils (CCT), liver, yolk sac, and spleen were aseptically cultured with tetrathionate enrichment broth. Enriched samples were confirmed negative for *Salmonella* by streak plating the samples on Xylose Lysine Tergitol-4 (XLT-4, Catalog no. 223410, BD Difco^TM^, Pittsburgh, PA, USA) selective media. Poults were placed in heated isolation cages with a controlled age-appropriate environment and provided with their own treatment diet and water ad libitum. On day 9 of age, all poults were orally challenged with 1 × 10^6^ CFU of ST per bird and weighed to calculate the concentration of fluorescein isothiocyanate-dextran (FITC-d) to be administered according to the group body weight. Subsequently, 24 h post-ST challenge (10 day-olds), poults were euthanized by CO_2_ inhalation, and CCT were collected aseptically to determine ST recovery as described below. Blood samples were collected from the femoral vein and centrifuged (1000× *g* for 15 min) to separate the serum for the determination of FITC-d, and intestinal samples (duodenum and ileum) were collected for morphometric analysis as described below.

### 2.8. Effect of Therapeutic Administration of FSBM on Salmonella typhimurium, Experiment 3

This experiment was performed to evaluate the therapeutic effect of FSBM in turkey poults infected with ST. One-day-old female turkey poults (Nicholas, Avigen turkeys) were obtained from a local hatchery. Poults were neck tagged, orally gavaged 1 × 10^4^ ST CFU/bird and randomly assigned to one of four different groups (*n* = 30 turkeys/group): (1) Control group, (2) FSBM group, (3) Control group challenged with ST (Control + ST) and (4) FSBM group challenged with ST (FSBM + ST). Poults were individually weighed at reception and on day 7 to obtain body weight (BW) and body weight gain (BWG). Poults were placed in heated isolation cages with a controlled age-appropriate environment and provided with their respective diet and water ad libitum. On day 6, all turkeys were weighed to calculate the concentration of FITC-d to be administered according to the group body weight. Subsequently, at d 7 days post-ST challenge, poults were euthanized, and CCT were collected aseptically to determine ST recovery as described below. Blood samples were collected from the femoral vein and centrifuged (1000× *g* for 15 min) to separate the serum for the determination of FITC-d and intestinal samples (duodenum and ileum) were collected for morphometric analysis as described below from ST challenged groups only, due to logistic issues.

### 2.9. Salmonella Recovery

CCT collected in experiments 2 and 3 were homogenized and diluted with saline (1:4 *w*/*v*), and ten-fold serial dilutions were plated on BGA supplemented with NO and NA, incubated at 37 °C for 24 h to enumerate total ST CFU. Following plating to enumerate total ST, the CCT samples were enriched in double strength tetrathionate enrichment broth and further incubated at 37 °C for 24. Enriched samples were streaked into XLT-4 selective media for confirmation of *Salmonella* presence. Plates that were negative on the enumeration method but were positive on enrichment were considered as 50 CFU/g as the limit of detection for ST viability.

### 2.10. Serum Determination of FITC-d Leakage

FITC-d (MW 3-5 KDa; Sigma-Aldrich, St. Louis, MO, USA) was used as a marker of paracellular transport and mucosal barrier dysfunction, as described by [16]. In both in vivo experiments, 1 h before the poults were humanely euthanized by CO_2_ inhalation, 24 turkey poults from each group were given an oral gavage dose of FITC-d (8.32 mg/kg of body weight).

### 2.11. Morphometric Measurement of the Small Intestinal Mucosa

Ileum and duodenum samples were collected (*n* = 6). A 1 cm segment of the midpoint of the duodenum and the distal end of the lower ileum from each bird was removed and fixed in 10% buffered formaldehyde for 48 h. Each of these intestinal segments was embedded in paraffin, and a 5-μm section of each sample was placed on a glass slide and stained with hematoxylin and eosin for examination under a light microscope. Morphometric analysis was performed as described previously [17]. All morphological parameters were measured using the ImageJ software package V1.53a (http://rsb.info.nih.gov/ij/). Six replicate measurements for each variable studied were taken from each sample, and the average values were used in statistical analysis. Villus length was measured from the top of the villus to the top of the lamina propria. Crypt depth was measured from the base upward to the region of transition between the crypt and villus. Villus width was measured at the widest area of each villus, whereas the villus:crypt ratio was determined as the ratio of villus height to crypt depth. Villus surface area was calculated using the formula (2π) (VW/2) (VL), where VW = villus width and VL = villus length [18].

### 2.12. Statistical Analysis

BW, BWG, Log CFU/g of ST, serum FITC-d concentrations, and intestinal morphometric measurements were subjected to analysis of variance (ANOVA) as a completely randomized design, using the general linear models procedure of SAS [19]. Significant differences among the means were determined by Duncan’s multiple range test at *p* < 0.05. The percent mortality and recovery of *Salmonella* were compared using the chi-square test of independence, testing all possible group combinations to determine significance for these studies.

## 3. Results

The results of the evaluation of the antimicrobial activity of the FSBM on ST using an in vitro digestive system in Experiment 1 are shown in Table 2. When FSBM replaced 100% of SBM in a turkey diet, a significant reduction (*p* < 0.05) in ST in the proventriculus and intestine compartments was observed when compared to the control group to undetectable levels (Table 2).

Table 3 summarizes the results of the evaluation of the FSBM on BW, BWG, the serum concentration of FITC-d levels, *Salmonella* recovery in CCT, and total mortality in turkey poults challenged with ST in the prophylactic model (Experiment 2) and therapeutic model (Experiment 3). In the prophylactic model, at day ten, a significant (*p* < 0.05) improvement in BW and BWG was observed in turkeys that were fed with the FSBM diets as compared to control groups. BW was significantly higher in the FSBM group (16.78 g) in comparison with the control group. After the ST challenge, the BW of control + ST was significantly (*p* < 0.05) reduced (28.46 g) as compared to FSBM + ST. The replacement of SBM for FSBM improved the BWG significantly by 11.73% in turkeys consuming the FSBM diet as compared to the control group. Similar results were observed in the BWG of poults that were challenged with ST were the replacement of the FSBM increased (*p* < 0.05) the BWG by 31.2% compared with control-ST turkeys (Table 3). Interestingly, in the prophylactic model, turkeys that received SBM, regardless of the ST challenge, showed a significant (*p* < 0.05) increase in serum levels of FITC-d when compared with turkeys poults that received the FSBM, with or without ST challenge. Nevertheless, turkeys that received FSBM and were challenged with ST presented with higher (*p* < 0.05) FITC-d levels than turkeys fed FSBM and received no ST challenge. In the prophylactic model, FSBM did not reduce ST recovery from CCT, and no mortality was recorded in any of the groups.

In the therapeutic model, a significant (*p* < 0.05) improvement in BW and BWG was observed in turkeys fed with the FSBM diets compared to control groups on day 7. BW was significantly (*p* < 0.05) higher in the FSBM group (14.71 g) in comparison with the control group. After seven days of ST challenge, BW of control + ST was significantly (*p* < 0.05) reduced (34.14 g) compared to FSBM + ST. The replacement of SBM for FSBM significantly (*p* < 0.05) improved the BWG by 17.55% in turkeys fed the FSBM diet as compared to the control group, or 145% between FSBM + ST group when compared to control + ST group (Table 3). In this model, turkeys that received the SBM diet also showed higher levels (*p* < 0.05) of FITC-d when compared with the turkeys fed with FSBM (Table 4). Although there were no significant differences in *Salmonella* recovery between treatments, turkeys fed with the FSBM and challenged with ST showed a substantial reduction (*p* < 0.05) in mortality by 25% when compared with control turkeys challenged with ST (Table 3).

The results of the evaluation of fermented soybean meal (FSBM) on the morphometric analysis of duodenum of turkey poults challenged with *Salmonella typhimurium* (ST) in the prophylactic model (Experiment 2) and therapeutic model (Experiment 3) are shown in Table 4. In the prophylactic model, at d 10, turkeys fed with FSBM diets without ST had a significant (*p* < 0.05) increase in villus length, villus surface area, and villus:crypt ratio when compared with the rest of the experimental groups. The Control group showed the lowest villus length and villus:crypt ratio, but higher villus width. No significant differences were observed between treatments that were challenged with ST in the prophylactic or therapeutic models (Table 4).

Table 5 summarizes the results of the evaluation of the FSBM on the morphometric analysis of the ileum of turkey poults challenged with ST in the prophylactic model (Experiment 2) and therapeutic model (Experiment 3). In the prophylactic model, at d 10, turkeys fed with the FSBM diet and challenged with ST showed a significant (*p* < 0.05) increase in the villus length, villus width, villus surface area, and villus:crypt compared with the rest of the experimental groups, and control group presented the lowest villus length and villus surface area (*p* < 0.05) when compared with turkeys that received the FSBM; however, crypt depth was significantly (*p* < 0.05) higher on turkeys that received FSBM, regardless of ST challenge (Table 5). In this study, no significant differences in the ileum mucosa were observed between treatments that were challenged with ST in both models (Table 5).

## 4. Discussion

Roughly ninety-eight percent of the world production of SBM is utilized as the main source of protein for livestock feed [20]. However, in monogastric animals, SBM has also been linked to a reduction in performance due to a number of anti-nutritional factors, including trypsin inhibitor, oligosaccharides, phytic acid, and allergenic proteins, such as β-conglycinin, and glycinin [21]. These anti-nutritional factors not only reduce the digestion, absorption, and utilization of nutrients but also induce gut inflammation, particularly in young animals [22]. Nevertheless, in recent years, numerous articles have shown that the fermentation of the SBM with beneficial bacteria and fungi can reduce these anti-nutritional factors present in SBM, improving performance and health in several monogastric animal models [6,13]. Furthermore, through fermentation processes, an increase of free amino acid content has been described [23]. In the present study, SBM was fermented with poultry origin strains of *L. salivarius* and *B. licheniformis*. Following fermentation, FSBM contained a total of 2 × 10^5^ LAB/g, 5 × 10^4^ total aerobic bacteria, and a total of 3 × 10^3^ spores/g. Both strains utilized for the fermentation process showed in vitro activity against ST, and this anti-bacterial activity was confirmed via the in vitro digestive model in the proventriculus and intestine compartments (Table 3). In addition, FSBM increases BW and BWG in the prophylactic and therapeutic models of ST infection (Table 3). In the prophylactic model, these improvements on performance were associated with significant changes in villus length, villus width, villus surface area, and villus: crypt ratio of the duodenum, and villus length, crypt depth, and villus surface area of the ileum (Table 4 and Table 5). Both findings observed in neonatal turkey poults are in agreement with previous publications using other monogastric animals [7,22,24].

The probiotic effect and reduction of enteric pathogens is another benefit of the inclusion of FSBM in animal diets [6,8,9,21]. However, in the present study, the complete substitution of SBM by FSBM, though shown in vitro activity against ST, did not reduce ST cecal colonization in either the prophylactic or therapeutic models. Even so, this outcome is not new in our laboratory’s experience translating in vitro data to in vivo efficacy. We have observed similar results with other probiotic bacteria against *Salmonella enteritidis* [25], or biodegradation of aflatoxin B1 [26], as well as other nutraceuticals showing significant and encouraging reductions against *Histomonas meleagridis* in vitro, but none of which translated to effectivity in vivo [27,28]. Hence, in vitro biological effects do not always draw a parallel result when tested in live animals. And yet, the FSBM significantly reduced the serum levels of FITC-d in both turkey models, regardless of whether they were challenged or not with ST, compared with turkey poults that received the SBM control diet (Table 3). Several studies conducted in our laboratory confirm that FITC-d in poultry is a good and reliable indicator to evaluate intestinal permeability utilizing several poultry models to induce intestinal inflammation [10,29]. These results suggest that FSBM improves intestinal integrity and reduces inflammation, as has been previously published [7,30]. It is worth mentioning that ST was highly pathogenic for neonatal turkeys poults, particularly in the therapeutic model, where 33.33% of the birds in the control + ST challenge group died, compared to the 8.33% mortality observed in the FSBM + ST birds (Table 3). The experiment had to be terminated three days earlier (day 7 of age) than initially planned because of the high morbidity and mortality observed in the challenge control group by day 6 of age. In further studies, a control containing a mixture of the SBM and the culture media without bacteria should be incorporated.

## 5. Conclusions

The fermentation of SBM with the LS and BL strains improved BW and BWG of neonatal turkey poults, and these changes were associated with morphometric changes in the intestinal mucosa (prophylactic model only), as well as an improvement in gut integrity by reducing leaky gut in turkeys challenged with ST in two experimental infective models. Although both strains used for the fermentation process showed in vitro activity against ST, no significant effect was observed in vivo. Nevertheless, FSBM reduced the severity of the ST infection in the therapeutic model. The fermentation with different beneficial bacteria and different inclusion rates of FSBM requires further investigation.

## Figures and Tables

**Table 1 animals-10-01849-t001:** Ingredients (%) of the diet used in all experiments.

Item Ingredients	Experimental Diets	
	Control %	FSBM %
Corn	55.53	55.53
Soybean meal	35.69	0
Fermented Soybean meal	0	35.69
Vegetable oil	4.22	4.22
Dicalcium phosphate	1.82	1.82
Calcium carbonate	1.12	1.12
Salt	0.38	0.38
DL-Methionine	0.37	0.37
Vitamin premix ^1^	0.20	0.20
L-Lysine HCl	0.28	0.28
Choline chloride 60%	0.20	0.20
Mineral premix ^2^	0.10	0.10
Selenium 0.6%	0.02	0.02
Propionic acid	0.02	0.02
Antioxidant ^3^	0.05	0.05
Total	100.00	100.00
**Chemical Analysis ^4^**		
ME (Kcal/Kg)	2800	2800
Crude Protein	27.81	28.35
Lysine	1.55	1.57
Methionine + Cysteine	1.018	1.00
Threonine	1.04	1.05
Fiber	2.70	2.72
Calcium	1.2	1.2
Nonphytate Phosphorus	0.6	0.6

FSBM = Fermented Soybean meal; ME = Metabolic energy; ^1^ Vitamin premix supplied the following per kg: vitamin A, 20,000,000 IU; vitamin D3, 6,000,000 IU; vitamin E, 75,000 IU; vitamin K3, 9 g; thiamine, 3 g; riboflavin, 8 g; pantothenic acid, 18 g; niacin, 60 g; pyridoxine, 5 g; folic acid, 2 g; biotin, 0.2 g; cyanocobalamin, 16 mg; and ascorbic acid, 200 g (Nutra Blend LLC, Neosho, MO 64850, USA). ^2^ Mineral premix supplied the following per kg: manganese, 120 g; zinc, 100 g; iron, 120 g; copper, 10–15 g; iodine, 0.7 g; selenium, 0.4 g; and cobalt, 0.2 g (Nutra Blend LLC, Neosho, MO 64850, USA). ^3^ Ethoxyquin; ^4^ Contents were analyzed on an as-fed basis. ME, Calcium, and Nonphytate Posphorus contents are calculated values.

**Table 2 animals-10-01849-t002:** Evaluation of the antimicrobial activity of fermented soybean meal (FSBM) on *Salmonella typhimurium* (ST) using an in vitro digestive system. Experiment 1.

Treatments	Log_10_ CFU/mL of ST In Vitro Digestive System		
	Crop	Proventriculus	Intestine
Control diet + ST	7.49 ± 0.11 ^a^	4.01 ± 0.03 ^a^	4.06 ± 0.06 ^a^
FSBM + ST	7.05 ± 0.04 ^a^	0 ± 0 ^b^	0 ± 0 ^b^

Data expressed as mean ± standard error (*n* = 15). Non-matching superscripts within a column differ significantly *p* < 0.05.

**Table 3 animals-10-01849-t003:** Evaluation of fermented soybean meal (FSBM) on body weight (BW), body weight gain (BWG), serum concentration of fluorescein isothiocyanate–dextran (FITC-d) levels, *Salmonella* recovery in ceca-cecal tonsils (CCT), and total mortality in turkey poults challenged with *Salmonella typhimurium* (ST) in the prophylactic model (Experiment 2) and therapeutic model (Experiment 3).

Treatments			Prophylactic Model ^1^ Experiment 2			
	BW at 10 Days (g)	BWG at 10 Days (g)	FITC-d (µg/mL)	CCT ST Log_10_ CFU/g ^2^	CCT +/− (%) ^3^	Total Mortality (%) ^4^
Control	195.22 ± 3.71 ^b^	132.48 ± 3.80 ^b^	205.00 ± 16.31 ^b^	0 ± 0	0/24 (0%)	0/24 (0%)
FSBM	212.00 ± 3.63 ^a^	148.03 ± 3.67 ^a^	74.73 ± 23.47 ^d^	0 ± 0	0/24 (0%)	0/24 (0%)
Control + ST	153.54 ± 4.66 ^d^	91.50 ± 4.70 ^d^	261.35 ± 19.36 ^a^	0.99 ± 0.34	8/24 (33.33%)	0/24 (0%)
FSBM + ST	182.00 ± 5.03 ^c^	120.12 ± 5.02 ^c^	167.61 ± 20.10 ^c^	0.13 ± 0.15	6/24 (25.00%)	0/24 (0%)
**Treatments**			**Therapeutic Model ^5^** **Experiment 3**			
	**BW at 7 Days (g)**	**BWG at 7 Days (g)**	**FITC-d (µg/mL)**	**CCT ST Log_10_ CFU/g ^2^**	**CCT +/− (%) ^3^**	**Total Mortality (%) ^4^**
Control	137.59 ± 2.70 ^b^	75.14 ± 2.70 ^b^	233.00 ± 26.31 ^a^	0 ± 0	0/24 (0%)	0/24 (0%)
FSBM	152.30 ± 2.34 ^a^	88.33 ± 2.51 ^a^	61.11 ± 16.98 ^b^	0 ± 0	0/24 (0%)	0/24 (0%)
Control + ST	85.86 ± 2.73 ^d^	24.50 ± 2.67 ^d^	218.62 ± 40.22 ^a^	3.75 ± 0.44	24/24 (100%)	8/24 (33.33%) ^a^
FSBM + ST	120.04 ± 3.30 ^c^	60.07 ± 3.08 ^c^	78.49 ± 16.98 ^b^	3.13 ± 0.38	24/24 (100%)	2/ 24 (8.33%) ^b^

^1^ Turkeys were orally gavaged with 10^6^ CFU/mL of *S. typhimurium* per turkey at 9-d old, and samples were collected 24 h later; ^2^ CFU data expressed as mean ± standard error (*n* = 24 turkeys); ^3^ Data expressed as positive/total turkeys (%); ^4^ Data expressed as death turkey poults/total turkey poults (%); ^5^ Turkeys were orally gavaged with 10^4^ CFU/mL of *S. typhimurium* per turkey at 1-d old, samples were collected seven days later. Superscripts within a column with no common superscript differ significantly *p* < 0.05.

**Table 4 animals-10-01849-t004:** Evaluation of fermented soybean meal (FSBM) on the morphometric analysis of the duodenum of turkey poults challenged with *Salmonella typhimurium* (ST) in the prophylactic model (Experiment 2) and therapeutic model (Experiment 3).

Treatments	Villus Length (µm)	Villus Width (µm)	Crypt Depth (µm)	Villus Surface Area (mm^2^)	Villus: Crypt Ratio (µm)
		**Prophylactic Model ^1^** **Experiment 2**			
Control	329.46 ± 9.10 ^c^	43.06 ± 1.71 ^a^	20.17 ± 0.71	44.75 ± 2.20 ^b^	17.03 ± 0.77 ^c^
FSBM	443.85 ± 9.59 ^a^	37.69 ± 1.42 ^b^	18.35 ± 0.46	52.79 ± 2.50 ^a^	24.69 ± 0.83 ^a^
Control + ST	381.67 ± 7.31 ^b^	35.45 ± 0.89 ^b^	20.54 ± 0.34	42.67 ± 1.14 ^b^	18.78 ± 0.49 ^cb^
FSBM + ST	392.12 ± 9.31 ^b^	35.65 ± 0.82 ^b^	20.49 ± 0.95	43.90 ± 1.45 ^b^	21.04 ± 1.36 ^b^
		**Therapeutic Model ^2^** **Experiment 3**			
Control	ND	ND	ND	ND	ND
FSBM	ND	ND	ND	ND	ND
Control + ST	338.50 ± 9.60	38.97 ± 1.45 ^a^	22.35 ± 1.07	41.44 ± 2.14	16.18 ± 0.76
FSBM + ST	371.22 ± 14.73	33.18 ± 1.00 ^b^	23.19 ± 0.69	38.37 ± 1.70	16.19 ± 0.54

^1^ Turkeys were orally gavaged with 10^6^ CFU/mL of *S. typhimurium* per turkey at 9-d old. Samples were collected 24 h later. ^2^ Turkeys were orally gavaged with 10^4^ CFU/mL of *S. typhimurium* per turkey at 1 d old. Samples were collected six days later. Data expressed as Mean ± standard error (*n* = 36 villus). Superscripts within a column with no common superscript differ significantly *p* < 0.05. ND = none determined.

**Table 5 animals-10-01849-t005:** Evaluation of fermented soybean meal (FSBM) on the morphometric analysis of the ileum of turkey poults challenged with *Salmonella typhimurium* (ST) in the prophylactic model (Experiment 2) and therapeutic model (Experiment 3).

Treatments	Villus Length (µm)	Villus Width (µm)	Crypt Depth (µm)	Villus Surface Area (mm^2^)	Villus: Crypt Ratio (µm)
		**Prophylactic Model ^1^** **Experiment 2**			
Control	112.27 ± 6.71 ^c^	25.45 ± 0.93 ^ab^	17.85 ± 0.89 ^b^	9.40 ± 0.84 ^c^	6.57 ± 0.41 ^b^
FSBM	165.83 ± 11.5 ^b^	24.52 ± 0.78 ^ab^	24.77 ± 0.98 ^a^	13.01 ± 1.13 ^b^	6.74 ± 0.40 ^b^
Control + ST	122.98 ± 6.83 ^c^	22.44 ± 0.83 ^b^	19.20 ± 1.02 ^b^	8.90 ± 0.65 ^c^	6.62 ± 0.37 ^b^
FSBM + ST	203.02 ± 5.20 ^a^	26.87 ± 1.05 ^a^	25.67 ± 0.99 ^a^	17.09± 7.64 ^a^	8.31 ± 0.37 ^a^
		**Therapeutic Model ^2^** **Experiment 3**			
Control	ND	ND	ND	ND	ND
FSBM	ND	ND	ND	ND	ND
Control + ST	151.76 ± 4.55 ^ab^	24.23 ± 0.71	23.78 ± 1.03 ^a^	11.70 ± 0.56 ^ab^	6.66 ± 0.28
FSBM + ST	130.21 ± 5.44 ^cb^	25.60 ± 0.71	23.53 ± 1.28 ^a^	10.59 ± 0.61 ^ab^	5.91 ± 0.30

^1^ Turkeys were orally gavaged with 10^6^ CFU/mL of *S. typhimurium* per turkey at 9 d old. Samples were collected 24 h later. ^2^ Turkeys were orally gavaged with 10^4^ CFU/mL of *S. typhimurium* per turkey at 1 d old. Samples were collected six days later. Data expressed as mean ± standard error (*n* = 36 Villus). Superscripts within a column with no common superscript differ significantly, *p* < 0.05. ND = none determined.

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
