# Peer review of "Assessment of Fermented Soybean Meal on Salmonella typhimurium Infection in Neonatal Turkey Poults"

_animals, 2020, doi:10.3390/ani10101849_

Round 1
Reviewer 1 Report
- L25 Replace both with two.
- L34 spell out PI
- L39 which strains?
- L45-47. Please rewrite. Salmonella spp. is not a foodborne pathogens because of consumption of undercooked food. Salmonella is a pathogen because of the virulence factors. Undercooked food could be free of pathogens.
- L57. What do the authors of intestinal integrity refer to?
- L87-88. Please clarify if these spores are from total microorganisms, only aerobic, or included anaerobic. It seems these are only from aerobic bacteria
- L90-93. It is not clear if the soybean meal was mixed with the two culture media and the two bacteria at the same time.
- L97-103. A control should be included containing a mixture of SBM+culture media without innocula.
- L180. For these and the other comparisons, a level of significance is needed.
The authors need to define if the FSBM includes the culture media of both bacteria.
Author Response
REVIEWER 1
Dear Reviewer, 1, thank you very much for the time you have spent on reviewing our manuscript your comments are very valuable and helpful for revising our paper and guiding our researches. We have studied those comments carefully and have made corrections, which we hope meet with the approval. Revised portion in the new version are highlighted in yellow.
The following is our point-by-point response to reviewers’ comments:
Comments and Suggestions for Authors
- L25 Replace both with two.
Suggestion accepted, thank you
- L34 spell out PI
Suggestion accepted, thank you
- L39 which strains?
Suggestion accepted, thank you
- L45-47. Please rewrite. Salmonella spp. is not a foodborne pathogens because of consumption of undercooked food. Salmonella is a pathogen because of the virulence factors. Undercooked food could be free of pathogens.
Suggestion accepted, text was modified, thank you
- L57. What do the authors of intestinal integrity refer to?
Suggestion accepted, text was modified, thank you
- L87-88. Please clarify if these spores are from total microorganisms, only aerobic, or included anaerobic. It seems these are only from aerobic bacteria
Suggestion accepted, text was modified, thank you
- L90-93. It is not clear if the soybean meal was mixed with the two culture media and the two bacteria at the same time.
Suggestion accepted, text was modified, thank you
- L97-103. A control should be included containing a mixture of SBM+culture media without innocula.
You are right about this, and unfortunately, we did not include such control, however, a statement has been included at the end of the discussion, thank you
- L180. For these and the other comparisons, a level of significance is needed.
Suggestion accepted, text was modified, thank you
The authors need to define if the FSBM includes the culture media of both bacteria.
Yes, this has been included, thank you
Reviewer 2 Report
Animals – 914857
The research is interesting, concerning the current problem of the use of refined protein feed and its influence on microbiological resistance of poultry. The manuscript is well written, but it requires a few improvements, which are listed below:
- Line 98-99 - please add information, according to which dietary recommendations the turkey feed was formulated; what kind of turkey strain was used in the experiment?
- Did fermentation affect the nutritional value of soybean meal? This information is needed when assessing the body weight gain of birds.
- Line 114, 134 - use the word "turkey" instead of "chicken"
- Table 1 – please provide the nutritional value of diets used in experiment
- Table 3-5 – please provide a P-value
Author Response
REVIEWER 2
Comments and Suggestions for Authors
The research is interesting, concerning the current problem of the use of refined protein feed and its influence on microbiological resistance of poultry. The manuscript is well written, but it requires a few improvements, which are listed below:
Dear Reviewer, 2, thank you very much for the time you have spent on reviewing our manuscript your comments are very valuable and helpful for revising our paper and guiding our researches. We have studied those comments carefully and have made corrections, which we hope meet with the approval. Revised portion in the new version are highlighted in yellow.
The following is our point-by-point response to reviewers’ comments:
- Line 98-99 - please add information, according to which dietary recommendations the turkey feed was formulated; what kind of turkey strain was used in the experiment?
Suggestion accepted, text was modified, thank you
- Did fermentation affect the nutritional value of soybean meal? This information is needed when assessing the body weight gain of birds.
Suggestion accepted, a chemical analysis has been incorporated into Table 1, that shows that the nutritional value of soybean meal was not affected in this trial, thank you
- Line 114, 134 - use the word "turkey" instead of "chicken"
Suggestion accepted, text was modified, thank you
- Table 1 – please provide the nutritional value of diets used in experiment
Suggestion accepted, Table 1 was modified, thank you
- Table 3-5 – please provide a P-value
Data in Tables 3-5 is expressed as mean ± standard error with P<0.05.
Reviewer 3 Report
The assessment of fermented feed on intestinal pathogens is a very interesting field. Efforts to improve farm animal health are an important contribution to animal welfare. Many thanks for your contribution.
Abbreviations should be defined in parentheses the first time they appear in the abstract, main text, and in figure or table captions and used consistently thereafter (see for example: line 34, PI; line 32, ST; line 35, FITC-d; line 69, SBM; line 89; etc.) Please check this for uniformity.
Line 25: the strains inhibited the in vitro activity against ST? I think you mean that they showed in vitro activity against ST.
Line 39: Were the strains used to initiate the fermentation process the same found in the fermentate at the time offered to the animals? Was this analyzed? If not, no statement can be done on the potential of this particular two bacterial strains to reduce ST in vivo. Even if fermentation was initiated with those strains, other strains might dominate the feed after the fermentation. Additionally, I am not sure if a simultaneous growth of the Bacillus and Lactobacillus strain is possible, or one inhibits growth of the other. It is of special interest to know which strains in which concentration were fed finally to the animals.
Line 42: italicize Salmonella
Line 47: leading cause is Campylobacter, you mean that within Salmonella these serovars are the most common. Nevertheless, the reference given was not an epidemiological study but evaluated detection methods! Please give the correct reference.
Line 49: I could not find the quoted statement in the results of the reference! Please check if the reference is the right one.
Line 53: there is only one reference concerning fermentation and nutritional value, reference number 5 did not evaluate fermentation process and intestinal inflammation
Line 56: reference?
Line 58: space between reference and the dot
In summary, I recommend to revise the introduction and references!
Line 75: abbreviation already mentioned above
Line 82, 107 and 161: even if already published I would appreciate a little description of the method
Line 94: Total count of lactic acid bacteria are not that high as expected in fermented feed, although high amounts of lactic acid bacteria were added before fermentation. Again, are there data concerning Bacillus counts? Do both strains perhaps inhibit the growth of each other? Are there data concerning lactic acid and pH? Those informations should be added to see if fermentation process was successful. Additionally, those data (pH, lactic acid concentration as well as counts of lactic acid bacteria) should be analysed and stated for the diets as fed.
Was the dry matter of the control and experimental diet comparable? I miss those values.
Table 2: The diet seem to have no effect on ST (indicated due to the results in the crop), why do the following compartments have effects? I miss values that describe the habitat, as pH for example.
Table 3. 10 or 9 day old chickens?
Tables 4 and 5: The values for the Control and FSBM groups are the same in Experiment 2 and 3. In the methods part it is not clear if both experiments share the same control groups. Whereas in Table 3, values are not the same between experiment 2 and 3 for Control and FSBM. You should check this.
As the chickens were orally gavaged at different ages even with different dosage, I would expect a different outcome of the infection; therefore, it is difficult to compare both trials (therapeutic and prophylactic model). It is a pitty that the age of dissection was different, as well as the duration of infection until dissection (but the cause was explained in the discussion). Still, what was the reason to choose a different age at infection?
Line 182f: grammar
Discussion
line 23: There are no data concerning ST counts in digesta, therefore a statement about intestinal colonization can not be done, more about translocation of bacteria in CCT
Line 35: there are no data collected that allow a statement with regard to a reduction in inflammation
Author Response
REVIEWER 3
The assessment of fermented feed on intestinal pathogens is a very interesting field. Efforts to improve farm animal health are an important contribution to animal welfare. Many thanks for your contribution.
Dear Reviewer, 3, thank you very much for the time you have spent on reviewing our manuscript your comments are very valuable and helpful for revising our paper and guiding our researches. We have studied those comments carefully and have made corrections, which we hope meet with the approval. Revised portion in the new version are highlighted in yellow. Furthermore, we are attaching some complementary articles for your consideration.
The following is our point-by-point response to reviewers’ comments:
Abbreviations should be defined in parentheses the first time they appear in the abstract, main text, and in figure or table captions and used consistently thereafter (see for example: line 34, PI; line 32, ST; line 35, FITC-d; line 69, SBM; line 89; etc.) Please check this for uniformity.
Suggestion accepted, text was modified accordingly, thank you
Line 25: the strains inhibited the in vitro activity against ST? I think you mean that they showed in vitro activity against ST.
Suggestion accepted, text was modified, thank you
Line 39: Were the strains used to initiate the fermentation process the same found in the fermentate at the time offered to the animals? Was this analyzed? If not, no statement can be done on the potential of this particular two bacterial strains to reduce ST in vivo. Even if fermentation was initiated with those strains, other strains might dominate the feed after the fermentation. Additionally, I am not sure if a simultaneous growth of the Bacillus and Lactobacillus strain is possible, or one inhibits growth of the other. It is of special interest to know which strains in which concentration were fed finally to the animals.
Before the fermentation process, microbiological analysis of the SBM was revealed to contain 1 x 103 colony forming units (CFU)/g of total lactic acid bacteria (LAB), and 1 x 104 CFU/g of total aerobic bacteria. A sample of the SBM was pasteurized at 70°C for 10 min to eliminate vegetative cells and validate the number of spores from aerobic bacteria per gram on samples plated on TS agar. The results showed that the SBM contained 1 x 102 spores/g. The fermented soybean meal (FSBM) was produced through fermentation of the soybean meal with L. salivarius and B. licheniformis according to methods presented in a previous study by Jazi et al. [11]. For the fermentation process, one kg of SBM was inoculated with the two-culture media and the two bacteria at the same time. Three liters of MRS containing 2 x 109 CFU/mL L. salivarius and three liters of TS broth containing 3 x 108 CFU/mL B. licheniformis and incubated for 24 h at 37°C, followed by additional 24 h at 30°C. FSBM was then dried in a hot-air oven at 60°C for 26 h. Following fermentation, samples from the FSBM were collected for determination of total LAB or total aerobic bacteria. A total of 2 x 105 LAB / g and 5 x 104 total aerobic bacteria were recovered from the FSBM. Pasteurization of the FSBM showed a total of 3 x 103 spores/g.
Line 42: italicize Salmonella
Suggestion accepted, text was modified, thank you
Line 47: leading cause is Campylobacter, you mean that within Salmonella these serovars are the most common. Nevertheless, the reference given was not an epidemiological study but evaluated detection methods! Please give the correct reference.
Suggestion accepted, text was modified, and three new references have been incorporated, thank you
Line 49: I could not find the quoted statement in the results of the reference! Please check if the reference is the right one.
Suggestion accepted, a new reference more a doc has been added, thank you.
Line 53: there is only one reference concerning fermentation and nutritional value, reference number 5 did not evaluate fermentation process and intestinal inflammation
Reference [5] Wang, W.; Wang, Y.; Hao, X.; Duan, Y.; Meng, Z.; An, X.; Qi, J. Dietary fermented soybean meal replacement alleviates diarrhea in weaned piglets challenged with enterotoxigenic Escherichia coli K88 by modulating inflammatory cytokine levels and cecal microbiota composition. BMC Vet. Res. 2020, 16, 245. doi: 10.1186/s12917-020-02466-5
Line 56: reference?
You are right, this is the first manuscript in which the strains used to ferment soybean meal are reported, hence the text was modified. Thank you
Line 58: space between reference and the dot
Suggestion accepted, text was modified, thank you
In summary, I recommend to revise the introduction and references!
Suggestion accepted, introduction and references were modified, thank you
Line 75: abbreviation already mentioned above
Suggestion accepted, text was modified, thank you
Line 82, 107 and 161: even if already published I would appreciate a little description of the method
Suggestion accepted, text was modified to describe each method, thank you
Line 94: Total count of lactic acid bacteria are not that high as expected in fermented feed, although high amounts of lactic acid bacteria were added before fermentation. Again, are there data concerning Bacillus counts? Do both strains perhaps inhibit the growth of each other? Are there data concerning lactic acid and pH? Those informations should be added to see if fermentation process was successful. Additionally, those data (pH, lactic acid concentration as well as counts of lactic acid bacteria) should be analysed and stated for the diets as fed.
You have a good point, pH and lactic acid concentration were not evaluated. However, counts of lactic acid bacteria (LAB), before and after SBM fermentation are reported in the manuscript. Inoculum of each bacteria was performed separately and added to the SBM as previously reported by Jazi et al. 2019. FSBM was dried in a hot-air oven at 60°C for 26 h., hence, it is expected to have such low numbers of LAB after those conditions, thank you
Was the dry matter of the control and experimental diet comparable? I miss those values.
Diet samples were chemically analyzed, amino acids were determined by ion chromatography (Biochrom Ltd. Cambridge, UK) (Table 1). Dry matter content of SBM and FSBM was 89.3% and 88.38%, respectively. This statement has been added to the text, thank you.
Table 2: The diet seem to have no effect on ST (indicated due to the results in the crop), why do the following compartments have effects? I miss values that describe the habitat, as pH for example.
That is an excellent question! We do not know and we wish we measured other variables of the habitat such as pH, thank you
Table 3. 10 or 9 day old chickens?
Suggestion accepted, text was modified, thank you
Tables 4 and 5: The values for the Control and FSBM groups are the same in Experiment 2 and 3. In the methods part it is not clear if both experiments share the same control groups. Whereas in Table 3, values are not the same between experiment 2 and 3 for Control and FSBM. You should check this.
Dear reviewer, with your invaluable observation, we realized we made a terrible mistake, thank you so much for catching it, it is remarkable that none of the co-authors or other two reviewers notice such important point. When we run the ANOVA, we included the values of both non-challenged groups from the prophylactic model into the therapeutic model, and this cannot be done, as the age of the turkeys is different. We have edited Tables 4 and 5 accordingly and make this important comment, as well as the respective changes throw all the manuscript. Thank you very much.
As the chickens were orally gavaged at different ages even with different dosage, I would expect a different outcome of the infection; therefore, it is difficult to compare both trials (therapeutic and prophylactic model). It is a pitty that the age of dissection was different, as well as the duration of infection until dissection (but the cause was explained in the discussion). Still, what was the reason to choose a different age at infection?
Indeed, we cannot compare the models as each model is completely different. We have published several articles using both models for several nutraceuticals. We are attaching three articles for your consideration. In our experience, there is a correlation between the age of the birds and the dose of Salmonella challenge. Hence, to obtain significant changes on intestinal colonization, different doses must use for each respective model. However, as we confirmed, turkey poults were extremely susceptible to the virulence of the ST we used for challenge. thank you
Line 182f: grammar
We did not find any grammar mistake on line 182, thank you.
Discussion
line 23: There are no data concerning ST counts in digesta, therefore a statement about intestinal colonization can not be done, more about translocation of bacteria in CCT
The word “intestinal” has been changed by “cecal”, thank you. Ceca-cecal tonsils (CCT) were collected in experiments 2 and 3 were homogenized and diluted with saline (1:4 w/v), and ten-fold serial dilutions were plated on BGA supplemented with NO and NA, incubated at 37°C for 24 h to enumerate total ST CFU.
Line 35: there are no data collected that allow a statement with regard to a reduction in inflammation
Several studies conducted in our laboratory confirm that FITC-d in poultry is a good and reliable indicator to evaluate intestinal permeability utilizing several poultry models to induce intestinal inflammation. FSBM significantly reduced the serum levels of FITC-d in both turkey models, regardless of whether they were challenged or not with ST, compared with turkey poults that received the SBM control diet (Table 3). We are attaching some manuscripts for your consideration, thank you.
https://www.ncbi.nlm.nih.gov/pmc/articles/PMC7362456/
https://www.ncbi.nlm.nih.gov/pmc/articles/PMC4388489/
https://pubmed.ncbi.nlm.nih.gov/31717681/
https://www.mdpi.com/2076-2615/9/4/184
https://www.frontiersin.org/articles/10.3389/fvets.2015.00066/full
https://www.frontiersin.org/articles/10.3389/fvets.2019.00282/full
Round 2
Reviewer 3 Report
Thank you for revising the manuscript according to my suggestions.
Just one futher suggestion: The description under table 3 further states that the animals were infected on day 10 of life. I think it should be day 9 as described in the text (line 152) and the other tables. Could you please check this?
Author Response
Dear Reviewer 3,
We have no words to express our gratitude for the time and work you have spent correcting and guiding us in this process. Your suggestions, clearly are priceless and have improved this manuscript in a remarkable way. The Editorial Board of Animals is very lucky to have reviewers of your caliber. Thank you so much!
Table 3 has been corrected, and challenge time now reads day 9. Thank you.
